# Towards a Newborn Screening Common Data Model: The Utah Newborn Screening Data Model

**DOI:** 10.3390/ijns7040070

**Published:** 2021-10-27

**Authors:** David Jones, Jianyin Shao, Heidi Wallis, Cody Johansen, Kim Hart, Marzia Pasquali, Ramkiran Gouripeddi, Andreas Rohrwasser

**Affiliations:** 1Centers for Disease Control and Prevention, Atlanta, GA 30333, USA; 2Utah Public Health Laboratory, Salt Lake City, UT 84129, USA; heidiwallis@utah.gov (H.W.); kimhart@utah.gov (K.H.); arohrwasser@utah.gov (A.R.); 3Unite Us, New York, NY 10007, USA; codyjohansen@gmail.com; 4Department of Pathology, School of Medicine, University of Utah and ARUP Laboratories, Salt Lake City, UT 84108, USA; pasquam@aruplab.com; 5Department of Biomedical Informatics, School of Medicine, University of Utah, Salt Lake City, UT 84108, USA; ram.gouripeddi@utah.edu

**Keywords:** newborn screening, newborn screening laboratory information management system, common data model, interoperability, electronic data exchange, NBS, LIMS, standards

## Abstract

As newborn screening programs transition from paper-based data exchange toward automated, electronic methods, significant data exchange challenges must be overcome. This article outlines a data model that maps newborn screening data elements associated with patient demographic information, birthing facilities, laboratories, result reporting, and follow-up care to the LOINC, SNOMED CT, ICD-10-CM, and HL7 healthcare standards. The described framework lays the foundation for the implementation of standardized electronic data exchange across newborn screening programs, leading to greater data interoperability. The use of this model can accelerate the implementation of electronic data exchange between healthcare providers and newborn screening programs, which would ultimately improve health outcomes for all newborns and standardize data exchange across programs.

## 1. Introduction

Newborn screening (NBS) increasingly utilizes electronic data to ensure the timely detection and treatment of a variety of endocrine, hematologic, immune, metabolic, and neurologic disorders. This increase in electronic data presents challenges for the exchange, processing, and storage of these data. To date, there is no common NBS data model. This situation is further complicated by the existence of differential data representation in databases that hold NBS-related information, including electronic health records (EHRs), laboratory information management systems (LIMS), and public health systems including vital records agencies. In the United States, NBS records are linked to the birth certificate and data from the Offices of Vital Records issuing birth certificates are frequently used to ensure quality of NBS demographic records. The Utah NBS Program has been faced with two challenges: (1) connecting and integrating the laboratory information management system (LIMS) directly into the health providers’ infrastructure to generate a complete chain-of-custody solution for the entire NBS life cycle, and (2) establishing a data warehouse structure to regularly generate and consume operational performance data across the NBS system.

With implementation requirements for a data warehouse, Laboratory Order Interface (LOI), and Laboratory Results Interface (LRI), the Utah NBS Program created a data dictionary that maps newborn screening data elements. Prior standardization work in NBS includes the establishment of NBS-specific Logical Observation Identifiers Names and Codes (LOINC) panels for NBS results and the development of implementation guides for the NBS LOI and LRI [1,2,3,4,5,6]. The identified challenges provided the opportunity to consolidate and expand on the data dictionary that the Utah NBS Program created and on prior work in the field to establish an NBS data model. Applying a patient-centric design, this NBS data model contains attributes of patient demographics, the associated guardian, associated primary care provider, associated submitter, specimen/s, factors that could impact the NBS, NBS laboratory orders, NBS laboratory results, confirmatory test orders, confirmatory test results, and final diagnoses.

Currently, the Utah NBS Program screens approximately 50,000 newborns annually. Utah is a two-screen state, performing newborn screening on specimens collected at 24 h–48 h of life and on specimens collected between 7 and 16 days of life. First-screen specimens are received from all 47 birth hospitals as well as from midwives serving the home-birth population. Second NBS specimens are predominantly received from pediatricians affiliated with major health networks as well as from private practice. Paper-based methods are used for both ordering and reporting NBS. Newborn demographic and related data are transcribed from EHR systems onto the NBS card that is used to collect the specimen. NBS results are sent to birth hospitals and primary care providers as a pdf report via direct email or electronic facsimile.

The Utah NBS program is transitioning from paper-based approaches toward automated, electronic methods to be able to harmonize patient data with EHRs used by birth hospitals, pediatricians, and clinical specialists. However, the implementation of interoperable data exchange protocols between EHRs and public health is difficult due to the existence of significant data challenges and limited informatics resources [7]. This article describes the development of a data model that can be used by the greater NBS community as a working foundation for the establishment of a consensus-based, common data model. This data model provides a standardized, consistent representation of required data to facilitate data exchange and process description, thereby accelerating electronic data exchange implementation and leading to overall process improvements. An improved ability to communicate data needs and requirements with birth hospitals, pediatricians, clinical specialists, and EHR vendors will ultimately improve health outcomes for all newborns. This article focuses exclusively on the standardization of such data elements; it does not describe the actual operational processes of the transactions nor the implementation procedures.

## 2. Methods and Results

Using a patient-centric design, the Utah NBS Program, ARUP Laboratories, major hospital networks in Utah, and the Utah Health Information Network staff developed an NBS data model that includes attributes such as patient demographics, associated guardian, associated primary care provider, associated submitter, specimen, factors that could impact the NBS, NBS laboratory orders, NBS laboratory results, confirmatory test orders, confirmatory test results, and final diagnoses. Representation from all process stakeholders ensured balanced model development. ARUP Laboratories represented the diagnostic reference laboratory performing confirmatory testing. To begin this process, the Utah NBS Program compiled the data collected on the NBS dried blood spot (DBS) collection device, data generated in the laboratory, data collected by reference laboratories performing second-tier and diagnostic testing, and data stored in the LIMS (confirmatory test results and diagnoses). Furthermore, desired information not currently being gathered was itemized and added to this list of data elements. The data elements were then sorted according to twelve data entities (Newborn Patient, Parent/Guardian, Submitter, Pediatrician, Factors that Impact NBS, Specimen/s, NBS Laboratory Order/s, NBS Laboratory Result/s, Confirmatory Testing Laboratory Order/s, Confirmatory Testing Laboratory Result/s, Diagnosis, and Post-NBS Treatment/s). Then, the 12 data entities were categorized into four NBS components (newborn patient, specimen/s, newborn screening, and follow-up) and relationships were determined among these data entities. Next, the data elements were mapped to the healthcare standards LOINC, SNOMED CT, and ICD-10-CM. This was followed by the mapping of the data elements to the HL7 v2.5.1 standard. As part of this mapping process, the most relevant answer list for the data elements and the HL7 data type were added to the NBS data model. In the final step, incoming messages at the Clinical Health Information Exchange (CHIE), Utah’s local health information exchange (HIE), were assessed to determine if data elements would be present in the admission, discharge, and transfer message (ADT); in the laboratory order message (OML^O21), and the observation result message (ORU^R01).

Table 1 lists the four NBS components and the twelve data entities of the data model. Each data entity contains different numbers of data elements. Some example data elements are provided in the table. For the complete data model, please see the Appendix A.

Figure 1 displays the conceptual structure of the proposed NBS Data model and the relationships among the NBS data entities in a logical representation schema.

The NBS component Newborn Patient contains five data entities: (1) Newborn Patient, (2) Parent/Guardian, (3) Submitter, (4) Pediatrician, and (5) Factors that Impact NBS. The Newborn Patient entity is associated with the Parent/Guardian, Submitter, and Pediatrician entities. Both the Newborn Patient and Parent/Guardian could be associated with Factors that Impact NBS: infant and/or maternal factors. Data for the Newborn Patient component originate from the birth hospital.

The NBS component Specimen/s contains one data entity: Specimen/s entity. Here, a Newborn Patient can be associated with one to many Specimen/s. Data for the Specimen/s component also originates from the birth hospital.

The NBS component Newborn Screening contains two entities: (1) NBS Laboratory Order/s and (2) NBS Laboratory Result/s; where Specimen/s can have NBS Laboratory Order/s and NBS Laboratory Order/s yield NBS Laboratory Result/s. NBS Laboratory Result/s may result in additional NBS Laboratory Order/s. Data for the newborn screening component originates from the NBS Program.

The NBS component Follow-Up contains four data entities associated with confirmatory testing, associated results, diagnoses, and treatment modalities: (1) Confirmatory Testing Laboratory Order/s, (2) Confirmatory Testing Laboratory Result/s, (3) Diagnosis, and (4) Post-NBS Treatment/s. NBS Laboratory Result/s may result in Confirmatory Testing Laboratory Order/s. Confirmatory Testing Laboratory Order/s have associated Specimen/s. Confirmatory Testing Laboratory Order/s yield Confirmatory Testing Laboratory Result/s. Confirmatory Testing Laboratory Result/s may result in additional Confirmatory Testing Laboratory Order/s. Confirmatory Testing Laboratory Result/s may or may not result in Diagnosis and Post-NBS Treatment/s. Data for the Follow-Up entity originate from diagnostic laboratories and clinical sub-specialists.

Post-NBS treatments are currently not tracked by the Utah newborn screening program. However, inclusion of this data entity in the data model is beneficial for the NBS community as it constitutes the foundation for long-term follow-up structures. Together with long-term care providers, we will develop and itemize the data elements associated with individual treatments.

Currently, the Utah NBS data model contains 648 data elements grouped into tables according to the respective data entities (see the Appendix A). During the development of the NBS data model, LOINC codes for spinal muscular atrophy (SMA) had not yet been established. To bridge this gap, LOINC codes were requested and have been approved and released for the SMA panel, determination, comment, and SMN1 measurement (cycle threshold). Evaluation and initial validation of the current Utah NBS data model consisted of discussion and reviews among the key stakeholders, the Utah NBS Program, ARUP Laboratories, major hospital networks in Utah, and Utah Health Information Network staff.

## 3. Discussion

An NBS common data model lays the system foundation and standard to describe and catalogue a spectrum of primary and derivatized data elements pertinent to NBS systems. For the respective programs, data elements allow for analytics-driven goal formulation, the prioritization of improvement opportunities, and ultimately, results in increased performance transparency and process improvement.

The developed NBS data model lays the foundation for the standardized implementation of electronic ordering and communication of NBS with all major healthcare networks in Utah. This implementation will eliminate paper-based ordering and reporting; it will eliminate transcription requirements from an EHR to the NBS collection device as well as from the collection device to the laboratory information management system, thereby significantly reducing rework requirements caused by transcription errors. It will also enable a complete chain-of-custody environment.

This data model provides the structure that would allow for the standardization of disparate NBS data generated by different screening programs in order to enable the global analyses and comparisons of systems. One such example of differential data representation is the date and time of birth. The Utah NBS Program, due to the historic design of the NBS card, did not collect time of birth, even though the information was available in the EHR. This can result in complications when attempting to accurately establish a complete chain of custody for a specimen and assess the timeliness of the entire NBS process. Likewise, not integrating the time of collection can interfere with result interpretation and reporting, especially when analyte concentrations undergo significant changes in the early newborn period.

We recommend this NBS data model for foundational use by other NBS programs as a standardized, structured methodology for the electronic exchange of information between EHRs and NBS programs. More specifically, the data model can serve as a guide for development of electronic orders, appropriate storage of data received/generated by an NBS program, and generation of outbound electronic results reporting. This NBS data model can serve as a working foundation for the creation of an NBS common data model.

From this point forward, the goal is to disseminate this model to the NBS community for feedback, review data elements currently captured by the NBS community, and perform gap analyses. These processes are intended to result in establishing a consensus-based NBS common data model and to promote its adoption across the NBS user community. An NBS common data model can provide the structure that would allow for the standardization of disparate NBS data generated by different screening programs to enable global analyses and comparisons of systems, ultimately leading to improved healthcare and health outcomes for newborns. Furthermore, an NBS common data model can be continually updated by the NBS community in order to account for the addition of new conditions. Such a model can also be structurally adapted to national standard modifications or requirements.

Previous efforts to standardize electronic messages in NBS have focused on leveraging OML^O21 and ORU^R01 for the exchange of patient demographic, laboratory orders, and laboratory results messages. Because previous approaches do not fully meet all of the requirements for a clinical follow up in the NBS process, the Utah NBS Program chose to map data to the ADT message. The ADT message is very useful for establishing patient identity in a LIMS for a majority of babies born in a state, allowing for the easier determination of missed newborn screens and the predictive forecasting of incoming sample volume. Furthermore, the ADT message can automatically receive patient updates such as diagnoses relevant to the NBS process and lays the foundation for efficient electronic long-term follow-up systems.

The findings in this report are subject to limitations. First, this data model was developed specifically for NBS using DBS; it does not include data elements for the screening of newborn hearing or critical congenital heart defect. However, the inclusion of these data elements could be simple to address due to the modular model design and existence of LOINC codes for these data elements. Next, whole genome sequencing and whole exome sequencing methods are increasingly adopted as second-tier screening strategies and confirmatory testing modalities for several of the new NBS disorders; however, they are not yet well-documented in the data model and, due to intrinsic complexity, are not included. Potential solutions include the use of Fast Healthcare Interoperability Resources (FHIR) [8]. Third, the data element for post-NBS treatments is still under development. This data entity entails long-term follow-up mechanisms and the involvement of long-term care specialists. Finally, an extensive evaluation of the model is required. Evaluation will entail further discussion among stakeholders, sharing the model online for continuous feedback (the purpose of this article); and surveying State NBS Programs using the system usability scale [9].

In summary, an NBS common data model promotes innovation within the NBS community’s health information technology systems and allows the NBS community to communicate data needs and requirements with birth hospitals, pediatricians, clinical specialists, and EHR and LIMS vendors. Standardized data representation solutions will further alleviate the dependence on monopolistic laboratory information system providers. The development of an NBS common data model can streamline the implementation of LOI and LRI, standardize the format of data storage within LIMS that would allow NBS Programs to modularize their LIMS, and allow for the development of standardized analytics tools that can be shared and applied across the NBS community. With the existence of an NBS common data model, data exchanges between NBS programs, public health agencies, and healthcare providers can become more meaningful, eventually resulting in improved health outcomes for all newborns.

## Figures and Tables

**Figure 1 IJNS-07-00070-f001:**
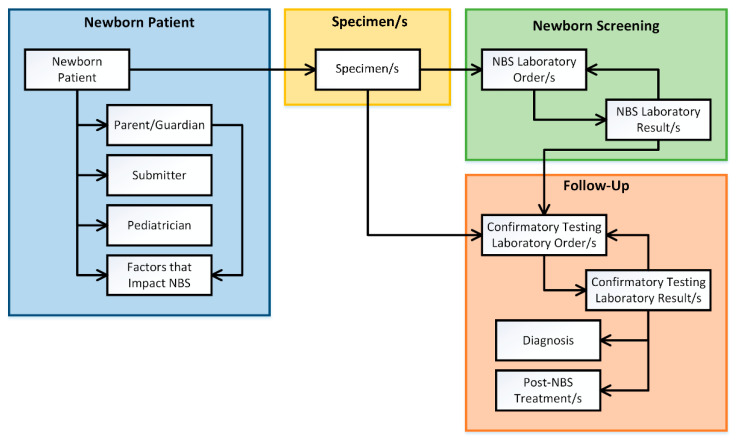
Logical representation of the newborn screening data model, displaying NBS data entities grouped by the NBS components and relationships between those NBS data entities. Abbreviations: newborn screening (NBS).

**Table 1 IJNS-07-00070-t001:** Example data elements of the newborn screening system components and the data entities.

NBS System Component	NBS Data Entity	Number of Data Elements	Example of Data Elements
Newborn Patient	Newborn Patient	11	Birth Date and Time, Birth Weight
	Parent/Guardian	9	Parent First Name, Parent Last Name, Parent Address
	Submitter	5	Submitter ID, Submitter Name
	Pediatrician	6	Pediatrician Name, Pediatrician Practice ID
	Factors that Impact NBS Interpretation	6	Feeding Type, Date of Transfusion
Specimen/s	Specimen/s	7	Specimen Collection Date/Time, Specimen Type
Newborn Screening	NBS Laboratory Order/s	25	Amino Acid Panel Lab Order, Cystic Fibrosis Panel Lab Order
	NBS Laboratory Result/s	236	Cystic Fibrosis Interpretation, Trypsinogen I Free Measurement
Follow-up	Confirmatory Testing Laboratory Order/s	18	Confirmatory Plasma Acylcarnitine Panel Order, Confirmatory Urine Amino Acids Panel Order
	Confirmatory Testing Laboratory Result/s	193	Confirmatory Plasma Acylcarnitine Profile Interpretation, Acetylcarnitine (C2) Measurement
	Diagnosis	132	Glucose-6-Phosphate Dehydrogenase Deficiency, Hemoglobin S Carrier
	Post-NBS Treatment/s	Under development	Required for implementation of long-term follow-up

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
