# Peer review of "Towards a Newborn Screening Common Data Model: The Utah Newborn Screening Data Model"

_2409-515X, 2021, doi:10.3390/ijns7040070_

Round 1
Reviewer 1 Report
Dear authors,
I highly recommend publishing your article in this journal. Your work is excellent and the paper is a high quality written. It is really innovative and it could be really useful for the most NBS programs around the world. It probably would be the first step to digitized NBS programs.
Here you have some comments and suggestions:
Introduction
The introduction lacks some detail in terms of background of NBS program in Utah. It is suggested that the introduction is expanded to include a few lines on paper-based approaches previously used (i.e. NBS cards, reports, etc), from how many birth hospitals are samples received, from how many newborns/year are samples received.
Methods and Results
Please, remove the discussion for the example mentioned (line 56-62) from this section. Some examples could be given, but the follow explanation should be in the Discussion Section. Please, modify the first paragraph to take this comment into account.
Also, when you say “Some NBS programs do not collect time of birth in their LIMS,…”, please use a cite or specify what NBS programs do it as example. It is not usual not to have this information.
DISCUSION
Some suggestions:
Please, specify who introduce each group of information with the Data Model system, i.e. “Newborn patient” data are introduced by the birth hospital.
In relation to the paperwork, please, specify what processes NBS program of Utah has been able to stop doing.
It would be useful mentioning the benefit of obtaining statistical data (request rate for 2nd samples, retest rate, unsatisfactory quality sample rate, etc.) in a very fast and agile way. Maybe, the environmental benefit too.
In line 94, please change SMN1 measurement for: SMN1 measurement (cycle threshold), “SMN1 measurement” or SMN1 gene analysis.
Figure 1
Please, check this information in your diagram:
- An additional arrow should be added from “NBS Laboratory Order/s” box to “Specimen” box, when you need a new sample?
- When you do a “Confirmatory Testing laboratory Order/s” you don’t usually use a DBS specimen. You don’t have samples used for confirmation (blood, urine, plasma) into “Specimen” group. Please, only review if an arrow from “Specimen” box to “Confirmatory Testing Laboratory Order/s” is correct.
The following comments are not necessary for the publication of the paper. They are suggestions to be in consideration on the table sent as supplementary material by authors (maybe are already included because there are a lot of data; if it is already included, skip it).
Specimen Analysis Data and Time
Specimen Validation Data and Time
Hours of life at the start of feeding
Types of transfusion (plasma / platelets / red blood cells / exchange transfusion).
Author Response
Thank you for reviewing our manuscript entitled Towards a Newborn Screening Common Data Model: The Utah Newborn Screening Data Model (IJNS-1351131). We appreciate the critical evaluation and have added to or revised the manuscript in response to your comments. Below is a list of the comments, our responses, and where the changes can be found in the manuscript.
Reviewer 1:
Point 1: The introduction lacks some detail in terms of background of NBS program in Utah. It is suggested that the introduction is expanded to include a few lines on paper-based approaches previously used (i.e. NBS cards, reports, etc), from how many birth hospitals are samples received, from how many newborns/year are samples received.
Response 1: We have added a paragraph providing the requested background information about the Utah NBS Program in the Introduction Section.
Point 2: Please, remove the discussion for the example mentioned (line 56-62) from this section. Some examples could be given, but the follow explanation should be in the Discussion Section. Please, modify the first paragraph to take this comment into account.
Response 2: We have moved the example to the Discussion Section.
Point 3: Also, when you say “Some NBS programs do not collect time of birth in their LIMS,…”, please use a cite or specify what NBS programs do it as example. It is not usual not to have this information.
Response 3: We have changed the sentence to indicate the Utah NBS Program does not collect time of birth due to historic design of the NBS card. This sentence is now in the Discussion Section.
Point 4: Please, specify who introduce each group of information with the Data Model system, i.e. “Newborn patient” data are introduced by the birth hospital.
Response 4: We have rewritten and expanded this section to improve the flow of the presentation. We clarified the Methods and Results Section specifying where the different data entities and their underlying data would originate. We have also included a table to showcase some sample data elements in each data entities.
Point 5: In relation to the paperwork, please, specify what processes NBS program of Utah has been able to stop doing.
Response 5: We have added a paragraph to the Discussion Section indicating that the Utah NBS Program is using the NBS data model to implement electronic ordering and reporting with several healthcare networks, thereby eliminating a majority of the paper-based methods and reducing work caused by transcription errors.
Point 6: It would be useful mentioning the benefit of obtaining statistical data (request rate for 2nd samples, retest rate, unsatisfactory quality sample rate, etc.) in a very fast and agile way. Maybe, the environmental benefit too.
Response 6: At this point in time, we are unsure of the need to include these calculated statistics in the data model where they are not currently electronically reported and do not have any associated standards (checked PHIN VADS). These data elements are exactly what the authors would like the community to identify and discuss for potential inclusion in an NBS common data model.
Point 7: In line 94, please change SMN1 measurement for: SMN1 measurement (cycle threshold), “SMN1 measurement” or SMN1 gene analysis.
Response 7: We have added “(cycle threshold)” to the Methods and Results Section as this was the code added after the gap was identified. The code/s for SMN1 and SMN2 gene analysis results already existed prior to the creation of this NBS data model.
Point 8: An additional arrow should be added from “NBS Laboratory Order/s” box to “Specimen” box, when you need a new sample?
Response 8: We have changed the description from Specimen to Specimen/s for both the NBS component and data entity to indicate that a patient can have one or many specimens. We also have modified the description of the figure to indicate the “Specimen/s” box does not refer to an actual specimen or specimens. It rather refers to a logical entity which can represent a variety of specimens using the different underlying data elements.
Point 9: When you do a “Confirmatory Testing laboratory Order/s” you don’t usually use a DBS specimen. You don’t have samples used for confirmation (blood, urine, plasma) into “Specimen” group. Please, only review if an arrow from “Specimen” box to “Confirmatory Testing Laboratory Order/s” is correct.
Response 9: Please see Response 8. The “Specimen/s” box represents a logical NBS data entity. As indicated in the newly added table, one of the data elements for “Specimen/s” is the specimen type. The data model describes a data entity rather than the actual specimen for testing. As such “Specimen/s” could entail e.g. the DBS card, a punch of the DBS card sent to a reference laboratory for second tier testing or a de novo blood collection used for diagnostic testing.

Reviewer 2 Report
Dear Authors:
I was VERY excited to see this article pop up in my in-box. I think this is important work and will shape the newborn screening world so thank you for your leadership on this topic. My biggest problem with the manuscript is an IT novice, I can't follow how my program could use this information. Spell it out for me. I also wanted to know more about your methods in terms of how decisions were made. Treat your process like a qualitative one - how were disagreements handled, did the whole group make a decision or just one person? You eventually list the stakeholder team but I'd do that early so I can assess if I think it is an appropriate level of expertise. Here are my specific questions:
- Who determined “further desired information not currently being gathered?” (lines 75-76)
- Who mapped the data elements? A person, a team? How did you solve any disagreements? (especially because you said “the most relevant answer” (line 80). Who decided what was most relevant? What was their expertise? Was it your stakeholders?
- What happens in the future as conditions are added? Or if race is recorded in a different way (e.g. Asian is broken out into Southeast Asian, East Asian, etc.)? How is this model updated?
- When I look at figure one, I see a list of attributes but I don’t know how it illustrates the common data elements. Can you show where HL7 comes in? CHIE, ADT, etc.?
- Or maybe you show a map of the data system prior to this process and then figure 1 to show the change.
- Maybe figure 1 should have a link to the supplemental materials? If not, a screen shot of one of the excel spreadsheet would help the reader a lot.
- Would everyone who wanted to build a system like this have to link to the LOINC, SNOMED, and ICD-10 codes or would we just used your list/mapping?
- I am not an HIT specialist so maybe this is obvious to your desired reader but . . . I do not understand how you get from the Excel sheet (supplemental) to your NBS data model. Do you only store the HL7 message which you show maps? Do you program your system to look for the LOINC, SNOMED, etc. codes and then write code so that the HL7 message is what is populated? How do you want me to use the information you present? I see your process but I do not fully understand what the end product looks like nor how my program might get there. Is your goal that I use your attributes?
- Can you add a section to your discussion on how newborn screening programs can use what you just presented?
I really like this idea and I know the field is very interested in seeing this work. If you could provide more detail, I think that would be helpful.
Author Response
Thank you for reviewing our manuscript entitled Towards a Newborn Screening Common Data Model: The Utah Newborn Screening Data Model (IJNS-1351131). We appreciate the critical evaluation and constructive comments. We have added to or revised the manuscript in response to your comments. Below is a list of the comments, our responses, and where the changes can be found in the manuscript.
Point 1: Who determined “further desired information not currently being gathered?” (lines 75-76)
Response 1: We have provided clarification in the Methods and Results Section to indicate that the NBS data model was evaluated by Utah NBS Program, ARUP Laboratories as the main provider of diagnostic testing services, major hospital networks in Utah, and Utah Health Information Network staff. We have added clarifying languages to the beginning of the Methods and Results Section indicating representatives involved in the development of the model. This expertise was selected to achieve representation from the respective system components.
Point 2: Who mapped the data elements? A person, a team? How did you solve any disagreements? (especially because you said “the most relevant answer” (line 80). Who decided what was most relevant? What was their expertise? Was it your stakeholders?
Response 2: Please see Response 1.
Point 3: What happens in the future as conditions are added? Or if race is recorded in a different way (e.g. Asian is broken out into Southeast Asian, East Asian, etc.)? How is this model updated?
Response 3: We have added clarification to the Discussion Section indicating this would be the purpose of the NBS community establishing an NBS common data model that can be continually updated by the community to address changes over time or to accommodate program specific requirements.
Point 4: When I look at figure one, I see a list of attributes but I don’t know how it illustrates the common data elements. Can you show where HL7 comes in? CHIE, ADT, etc.?
Response 4: We agree with the reviewer and have expanded and rewritten this section. We have also provided a table with the components, data entities as well as examples data elements to clarify the model. We have provided clarification describing Figure 1 to the Methods and Results Section and clarified the caption for Figure 1.
Figure 1 is a logical representation of the NBS data model, this is a commonly developed figure for representing data models that displays the high-level entities, data elements contained within those data entities, and relationships between them.
The data elements and all associated mappings are present within the tables which are available as supplemental material due to the volume of data elements. In the mappings, the data elements were mapped to different types and respective segments of HL7 messages as well as appropriate codes.
Point 5: Or maybe you show a map of the data system prior to this process and then figure 1 to show the change.
Response 5: Please see Response 4. While this is a logical and excellent suggestion, prior to this work, a comprehensive data model did not exist. This lack prompted the development of such conceptual model that ultimately guided the transition of a new LIMS and the interoperability framework for Utah. For clarification, we also have included a table to showcase some example data elements of each data entity in the data model.
Point 6: Maybe figure 1 should have a link to the supplemental materials? If not, a screen shot of one of the excel spreadsheet would help the reader a lot.
Response 6: Please see Response 4 and 5.
Point 7: Would everyone who wanted to build a system like this have to link to the LOINC, SNOMED, and ICD-10 codes or would we just used your list/mapping?
Response 7: We have added a description to the Discussion Section for how NBS Programs can use the NBS data model. We emphasize the common data model as a resource to guide program specific development.
Point 8: I am not an HIT specialist so maybe this is obvious to your desired reader but . . . I do not understand how you get from the Excel sheet (supplemental) to your NBS data model. Do you only store the HL7 message which you show maps? Do you program your system to look for the LOINC, SNOMED, etc. codes and then write code so that the HL7 message is what is populated? How do you want me to use the information you present? I see your process but I do not fully understand what the end product looks like nor how my program might get there. Is your goal that I use your attributes?
Response 8: Conceptually, an NBS data model should contain all data elements relevant to NBS and the relationships among them. As we now clarified in the paper, the Utah Newborn Screening Program used this data model for guidance to provide a standard structure of electronic information exchange among our partners: major hospital networks, Utah Health Information Network and external laboratories or service providers. In practice, we exchange information via HL7 v2.5.1, a commonly used standard. Thus, the data elements are mapped to the appropriate segments of different HL7 messages and/or applicable codes. Not all data elements in this model are used in our implementation due to various practical reasons. We specify the minimal required data elements and some optional but desirable data elements so our partners know how to construct or parse the HL7 messages according to the data model and data relationships.
In the paper, we emphasize our goal to disseminate this data model to the NBS community for review and feedback and to fill missing gaps. An NBS common data model with a broader consensus will facilitate the adoption of such data model and standardize the implementation of electronic exchange of NBS information.
Point 9: Can you add a section to your discussion on how newborn screening programs can use what you just presented?
Response 9: Please see Response 7 and 8. We throughout the manuscript differentiated data model from implementation models and processes. We now also emphasize that such model can use as a foundation for program specific implementation requirements.
Round 2
Reviewer 2 Report
I appreciated the authors' response to my questions. I feel the paper is much clearer. I especially appreciate Table 1. Again, I think what you have done is so important to the NBS community and I am excited to use the work you have done to strengthen the NBS work I do. Thank you.